A survey on surface reconstruction based on 3D Gaussian splatting

Xu Zheng
Chen Gang chengang_vge@163.com
Li Feng
Chen Lingyu
Cheng Yuanhang
Institute of Geographical Spatial Information, Information Engineering University , Zhengzhou , China
Coelho Paulo Jorge
Electronic publication date: 2025 Aug 5
Publication date: 2025
Volume: 11
Electronic Location ID: e3034
Received 2025 Jan 6; Accepted 2025 Jun 24
Copyright: © 2025 Xu et al.
Copyright year: 2025
Copyright holder: Xu et al.
License: This is an open access article distributed under the terms of the Creative Commons Attribution License, which permits unrestricted use, distribution, reproduction and adaptation in any medium and for any purpose provided that it is properly attributed. For attribution, the original author(s), title, publication source (PeerJ Computer Science) and either DOI or URL of the article must be cited.
License URL: https://creativecommons.org/licenses/by/4.0/

Keywords: 3D reconstruction, 3D Gaussian splatting, Surface reconstruction, Multi-view imagery

Funding: Key Project of the National Natural Science Foundation of China 42130112 This work was supported by the Key Project of the National Natural Science Foundation of China (Grant No. 42130112). The funders had no role in study design, data collection and analysis, decision to publish, or preparation of the manuscript.

==============================
Surface reconstruction is a foundational topic in computer graphics and has gained substantial research interest in recent years. With the emergence of advanced neural radiance fields (NeRFs) and 3D Gaussian splatting (3D GS), numerous innovative many novel algorithms for 3D model surface reconstruction have been developed. The rapid expansion of this field presents challenges in tracking ongoing advancements. This survey aims to present core methodologies for the surface reconstruction of 3D models and establish a structured roadmap that encompasses 3D representations, reconstruction methods, datasets, and related applications. Specifically, we introduce 3D representations using 3D Gaussians as the central framework. Additionally, we provide a comprehensive overview of the rapidly evolving surface reconstruction methods based on 3D Gaussian splatting. We categorize the primary phases of surface reconstruction algorithms for 3D models into scene representation, Gaussian optimization, and surface structure extraction. Finally, we review the available datasets, applications, and challenges and suggest potential future research directions in this domain. Through this survey, we aim to provide valuable resources that support and inspire researchers in the field, fostering advancements in 3D reconstruction technologies.

Introduction

Three-dimensional (3D) reconstruction refers to the process of recreating various aspects of a real-world visual environment in a virtual space using techniques such as computer vision and image processing. These aspects include the geometry of objects, the motion of specific elements within a scene, and the observed textures and appearances (Ingale & Jadhav, 2021). Geometric reconstruction of objects in real scenes plays a critical role in domains such as digital urban modeling, virtual geographic environments, and ecological protection.

The advent of neural radiance fields (NeRF) (Mildenhall et al., 2022) marks a significant milestone in computer graphics and 3D reconstruction. NeRF has revolutionized novel view synthesis and introduced a new paradigm for generating 3D model surfaces.

NeRF and related methods have incorporated techniques such as occupancy networks (Oechsle et al., 2021), signed distance functions (SDFs) (Yariv et al., 2021), and coordinate-based multilayer perceptrons (MLPs) (Reiser et al., 2021; Fridovich-Keil et al., 2022) for surface reconstruction tasks. While these methods generate smooth and continuous surfaces, they often struggle to capture high-fidelity shape details and typically exhibit slow optimization processes. To address these limitations, one research direction reduces reliance on MLP layers by decomposing scene information into separable structures such as points (Xu et al., 2022) or voxels (Liu et al., 2020a; Li et al., 2023b), thereby enhancing computational efficiency. Another approach focuses on minimizing geometric errors through methods such as weighted volume rendering (Wang, Liu & Yang, 2021) or by decomposing implicit SDFs (Wang, Skorokhodov & Wonka, 2022) into basis and displacement functions to progressively optimize surface geometry for high-frequency details. To further improve reconstruction quality, some studies (Fridovich-Keil et al., 2022; Müller et al., 2022; Sun, Sun & Chen, 2022) employ hash table encoding or combine surface and volume rendering (Reiser et al., 2024; Turki et al., 2024) to boost rendering speed and accuracy. Although NeRF has made substantial progress in modeling precision, it continues to face challenges in computational efficiency and controllability. In this context, the introduction of 3D Gaussian splatting (3D GS) (Kerbl et al., 2023) has redefined scene representations and significantly improved rendering performance.

Even before the emergence of NeRF, substantial advancements had been made in 3D geometric modeling. Traditional methods for 3D reconstruction based on point clouds (Botsch et al., 2005; Yifan et al., 2019) and meshes (Munkberg et al., 2022) combined with modern GPU rasterization techniques, offered faster processing speeds but often suffered from rendering artifacts and discontinuities. Multi-view stereo techniques (Goesele et al., 2007) estimate depth using feature matching (Schönberger et al., 2016) or employ voxel-based shape representations (Seitz & Dyer, 1999; Tulsiani et al., 2022) to generate point clouds or voxel grids. However, the resolution limitations of these outputs restrict the accuracy of reconstructed models. Deep learning-based mesh modeling (Wang et al., 2018; Liu et al., 2020b; Riegler & Koltun, 2020; Riegler & Koltun, 2021) integrates graphics with neural networks to enhance rendering quality. Although these approaches utilize computational resources efficiently, they often struggle to produce 3D models with precise geometric and textural fidelity in complex scenes.

Compared to traditional oblique photogrammetry techniques (Cramer et al., 2018; Przybilla et al., 2019), 3D GS approaches have demonstrated exceptional performance when dealing with weak textures, fine structures, and non-Lambertian surfaces. These methods enable photorealistic 3D reconstructions characterized by rapid data acquisition, robust scene recovery, and high modeling accuracy. Integrating 3D GS with rasterization techniques has resulted in faster rendering speeds than those achieved by NeRF. Models represented using 3D Gaussians often require targeted operations to refine their geometry. For example, combining 3D Gaussians with implicit neural surfaces (Chen, Bao & Wang, 2023; Yu et al., 2024b) extraction while applying regularization constraints (Chen, Huang & Zhao, 2024; Fan et al., 2024) helps align Gaussians more closely with model surfaces. Additionally, deformable Gaussians (Song et al., 2024) describe model motion, facilitating surface shape extraction in dynamic scenes.

To help readers keep pace with the rapid advancements in 3D GS, this survey provides a comprehensive and current review of surface reconstruction techniques based on 3D GS, highlighting recent achievements in the field. As an emerging technology, 3D GS has experienced significant growth in surface reconstruction applications. This survey systematically examines these developments and their diverse contributions. We offer a structured overview and roadmap of recent studies, focusing on scene representation, reconstruction methods, datasets, and applications while emphasizing the remaining challenges.

This survey specifically addresses surface reconstruction methods using 3D GS, along with related datasets and applications. A structured outline of the article is shown in Fig. 1 and organized as follows: “Background” introduces the foundational concepts of 3D GS and related research. “Key Technologies” discusses key processes, including scene representation, 3D Gaussian optimization, and surface structure extraction. “Datasets” reviews widely used datasets, and “Applications” highlights representative applications such as autonomous driving, digital human modeling, large-scale scene construction, and 3D editing. “Challenges” and “Conclusions” examine open challenges and conclude the survey with a balanced perspective on the current state and future outlook of 3D GS in surface reconstruction.

Figure 1 Overview of this survey, including 3D scene representation, surface reconstruction, datasets, and applications.

Background

3D GS is an advanced modeling and visualization technique that enables real-time rendering of high-resolution images. This section provides a foundational overview of 3D GS. 3D Gaussian Splatting introduces the construction of well-defined 3D Gaussians for image processing. Contextual Concepts reviews related technologies and scientific developments in the field. Survey Methodology outlines the approach used for the literature search.

3D Gaussian splatting

A 3D Gaussian is defined by parameters including position, opacity, a 3D covariance matrix, and color. All of these parameters are learnable and continuously optimized through backpropagation. The position of a 3D Gaussian can be represented as:

(1) G(x)=e−12xT∑−1x

where ∑ is the covariance matrix that controls the size and orientation of the 3D Gaussian distribution.

During the rendering process, the 3D Gaussian is first projected from the 3D space onto a 2D Gaussian. Given an observation transformation W and a 3D covariance matrix ∑, the corresponding 2D covariance matrix ∑′ is computed using a projection matrix

(2) ∑′=JW∑WTJT

where W is the observation transformation matrix, and J is the Jacobian matrix obtained from the affine approximation of the projection transformation. Subsequently, the Gaussians within the same pixel block are sorted, and alpha blending is applied to these 2D Gaussians to compute the color of each pixel.

(3) G(x′)=∑i∈Nciσi∏j=1i−1(1−σj),σi=αiGi′(x′)

where x′ is the pixel position on the image plane, and αi is the opacity of the i-th Gaussian.

Contextual concepts

Several techniques and research areas are closely related to 3D GS and are briefly introduced below.

Surface reconstruction

Surface reconstruction is a foundational area in computer graphics and vision, aiming to generate complex and accurate surfaces from sparse input data (Chen, Huang & Zhao, 2024). Multi-view-based surface reconstruction is a key component of 3D reconstruction and can be classified into traditional surface reconstruction, neural surface reconstruction, and 3D Gaussian-based surface reconstruction, depending on the intermediate representations and processing methodologies used.

Point-based representations

Point-based methods (Kobbelt & Botsch, 2004) offer topologically flexible, efficient, and adaptable representations of 3D model geometry (Insafutdinov & Dosovitskiy, 2018; Wiles et al., 2020; Aliev et al., 2020). These methods are notable for their fast rendering speeds and high editability (Zwicker et al., 2001a; Lassner & Zollhöfer, 2021). Points may be rendered directly using a fixed size or converted into simple primitives such as spheres, ellipsoids, or surfaces (Pfister et al., 2000; Zwicker et al., 2001b; Botsch et al., 2005), which are then projected onto the imaging plane to enhance rendering quality.

Survey methodology

The aim of this survey is to systematically summarize surface reconstruction methods based on 3D GS, along with relevant datasets and applications. Due to the novelty of this technology, we consulted three electronic databases: the Computer Vision Foundation Open Access Database, arXiv.org, and Web of Science. A comprehensive literature search was conducted using the keywords 3D Gaussian splatting, 3D GS, mesh, and surface. We systematically assess the collected articles by considering the innovativeness and representativeness of scene representation, optimization strategies, and surface extraction methods, using these criteria to guide the selection and filtering of articles. Although it is challenging to cover all the methods related to surface reconstruction based on 3D GS, we have tried our best to include the main methods and provided comprehensive descriptions.

Key technologies

In 3D reconstruction, selecting an appropriate model for scene representation is essential. Leveraging multi-view imagery, 3D GS enables high-fidelity geometric reconstruction and rendering quality. In the following sections, we outline the typical steps involved in surface reconstruction methods based on 3D GS. We begin with scene representation “Scene Representation”, then describe the optimization of 3D Gaussians for a given scene “Optimization Process of 3D Gaussians”, and finally explore methods for extracting surface structures “Surface Structure Extraction”.

Scene representation

Accurately fitting 3D Gaussians to the model surface is a critical step when initializing a scene using 3D GS. The input for 3D GS includes multi-view imagery and camera pose data, while the output is typically a mesh or point cloud. This section outlines the key characteristics and motivations behind surface reconstruction techniques based on 3D GS. Depending on whether the target scenes involve motion, they can be categorized as static or dynamic. For static scenes, factors such as model size, quantity, and geometric complexity must be considered. In contrast, for dynamic scenes, modeling deformation over time introduces additional complexity, making surface structure extraction significantly more challenging.

Static scene representation

Implicit neural surfaces or explicit primitives are commonly used to represent static scenes. In neural representations of 3D surfaces, occupancy networks (Niemeyer et al., 2020) and SDF (Yariv et al., 2020) are often combined with neural volume rendering techniques (Mildenhall et al., 2022), yielding strong results in surface prediction and novel view synthesis. To reconstruct object-centric or large-scale indoor and outdoor scenes, Neuralangelo (Li et al., 2023b) integrates multiresolution hash grids with neural surface rendering and employs multiresolution hash encoding to improve the capacity of neural SDFs in representing surface contours of 3D models. NeuSG (Chen, Bao & Wang, 2023) uses a neural implicit model to predict a surface-derived normal prior to refining the 3D Gaussian point cloud rather than directly generating point clouds from multi-view imagery. It introduces a scale regularizer that compels 3D Gaussians to deform into thin structures, approximating the model surface. Recognizing that constraining the density field and primitive shape may degrade rendering quality, Gaussian Splatting Distance Fields (GSDF) (Yu et al., 2024b) employs SDFs to guide density control in 3D Gaussians, producing tightly distributed structures on the model surface. This method combines the complementary strengths of SDFs and 3D Gaussians for scene representation, reducing floating artifacts and enhancing edge detail in synthesized views, which results in high-fidelity geometric reconstruction. However, this approach may compromise surface smoothness when handling complex texture patterns.

In explicit representation methods, surface elements (Pfister et al., 2000) have been shown to effectively represent complex geometric structures. Because 3D Gaussians may struggle to align closely with the model surface due to multi-view inconsistencies, some approaches represent 3D Gaussians as 2D planes. Early studies using 2D Gaussians for surface reconstruction required dense point clouds or precise surface normals as auxiliary data (Kopanas et al., 2021). However, 2D Gaussian Splatting (2DGS) (Huang et al., 2024a) models 3D Gaussians as 2D Gaussian discs. In this approach, density is distributed within the Gaussian disc, and the normal is defined as the direction of the steepest density change, enabling better alignment between the discs and the object surface. Quadratic Gaussian Splatting (QGS) (Zhang et al., 2024b) employs quadric surfaces to represent 3D Gaussians, defining Gaussian distributions on quadric paraboloids. This representation enables continuous conversion between convex and concave forms, improving geometric fitting and allowing for more accurate capture of complex surface textures. Recognizing that vehicle trajectories are typically parallel to road surfaces, RoGS (Feng et al., 2024) initializes the coordinates of Gaussian planes using trajectory point data to ensure that the 2D Gaussian Road surface closely matches the actual road. Beyond converting 3D Gaussians into 2D planes, triangular meshes can also be used to represent a model’s geometric structure, offering high rendering quality and supporting mesh manipulation and 3D editing of the model’s appearance and shape. For instance, Mani-GS (Gao et al., 2024a) extracts triangular meshes using neural surface fields, binds 3D Gaussians to triangle primitives, and manipulates them through mesh operations to enable deformation and soft-body simulation. TetSphere (Guo, Tang & Chen, 2024) represents model geometry using tetrahedral grids, initializing a set of tetrahedra that can generate arbitrary topologies by adjusting vertex positions to match scene geometry. While representing 3D Gaussians with alternative structures improves geometric fidelity, it may reduce computational efficiency and rendering performance. RaDe-GS (Zhang et al., 2024a) addresses this limitation by defining ray intersections on the model surface and treating them as Gaussian maxima, significantly improving surface fitting accuracy while maintaining rendering efficiency. Mip-splatting (Yu et al., 2024a) applies 2D Mip filters and 3D smoothing filters to constrain the maximum frequency of the 3D Gaussian representation, thereby limiting Gaussian size and reducing aliasing artifacts in the scene.

In addition, combining multi-view imagery with sensor data is an important approach to scene representation. Snap-it (Comi, Sarti & Perotti, 2024) integrates touch data with multi-view imagery, reconstructing glossy or reflective object geometries by reducing the opacity at touch locations, and achieves scene reconstruction speeds several times faster than existing methods. Tightly Coupled LiDAR-Camera Gaussian Splatting (TCLC-GS) (Zhao, Huang & Fang, 2024) proposes a method that combines Light Detection and Ranging (LiDAR) sensors with cameras, using colored 3D meshes and implicit hierarchical octrees to complete scene representation and produce high-quality 3D reconstructions. SplaTAM (Keetha et al., 2024) is the first to combine 3D GS with Red-Green-Blue-Depth (RGB-D) cameras, modeling the world as a collection of 3D Gaussian distributions. This approach demonstrates state-of-the-art performance in camera pose estimation, scene reconstruction, and novel view synthesis. Gaussian splatting-Simultaneous Localization and Mapping (GSSLAM) (Matsuki et al., 2024) employs random sampling of depth rendering for 3D Gaussian position distribution and proposes an analytical Jacobian matrix for optimizing camera pose, along with a regularization method based on Gaussian shape, to achieve real-time, high-fidelity, and wide-range localization. It performs particularly well in reconstructing transparent and small objects. Gaussian-SLAM (Yugay et al., 2023) divides the scene into sub-images and proposes effective strategies for efficient seeding of 3D Gaussians and online optimization, thereby enabling 3D reconstruction in large-scale environments.

Among these approaches, implicit representation methods can recover the geometric details of 3D models within a scene and produce superior reconstruction results, although Signed Distance Functions (SDFs) tend to struggle with reconstructing transparent objects. Explicit-primitive-based methods require carefully designed constraints tailored to the scene to ensure that 3D Gaussians adhere closely to the model geometry. For static scenes, maximizing the rendering efficiency of 3D models while preserving modeling accuracy is essential. Achieving this balance is a key consideration in the design of scene representation methods.

Dynamic scene representation

In dynamic scenes, representing object deformations using 3D Gaussians remains a significant challenge. GSDeformer (Huang & Yu, 2024) addresses this by converting 3D Gaussians into proxy point clouds and applying cage-based deformation to manipulate the point cloud, thereby deforming the moving model. Scale-adaptive Gaussian splatting (SA-GS) (Song et al., 2024) and Sparse-controlled Gaussian splatting (SC-GS) (Huang et al., 2024b) propose novel scene representation methods that explicitly separate motion and appearance into sparse control points and dense Gaussians. The sparse control points are used to learn a 6-degree-of-freedom transformation basis, with local interpolation weights used to generate a 3D Gaussian motion field that reconstructs scene appearance, geometry, and motion. DG-Mesh (Liu, Su & Wang, 2024) extracts a mesh from 3D Gaussian points and tracks vertex changes over time to construct a temporally consistent mesh sequence that captures complex surface structures. MaGS (Ma, Liu & Wu, 2024) introduces a learnable relative deformation field to model the displacement between a mesh and 3D Gaussians, accurately capturing Gaussian motion and depicting realistic dynamic object deformation. Interactive (Zhan, Zhang & Liu, 2024) decouples material and lighting information from a single human video and uses SDFs to construct a regularized volumetric mesh. Each vertex is assigned an attribute, and attributes of the 3D Gaussians in canonical space are interpolated from nearby mesh vertices to represent a dynamic, interactive digital human. MoSca (Lei, Wu & He, 2024) decouples geometric information from the deformation field and models motion using a motion scaffold deformation representation.

The use of discrete 3D Gaussians to represent dynamic 3D models enables the capture of fine motion details, supporting rapid training and real-time rendering. This capability is a key consideration in the design of dynamic scene representations. However, training a large number of 3D Gaussians remains a major challenge for computing hardware. Consequently, achieving efficient training under limited computational resources is a central issue in this field. Future research should explore strategies to use fewer 3D Gaussians while maintaining the ability to represent complex scene models.

Optimization process of 3D Gaussians

Due to the substantial differences between the rendered images of an initialized scene and real-world images, 3D Gaussians must be optimized during training to ensure accurate surface fitting to the 3D model. During this process, it is often necessary to consider the attributes of 3D Gaussians and incorporate regularization constraints into the loss function to guide learning in the desired direction. In practice, additional constraints are frequently designed to account for the specific characteristics of a given scene. Table 1 summarizes commonly used optimization constraints for geometric surface reconstruction based on 3D GS.

Table 1 Optimization constraints for surface reconstruction of 3D Gaussians.

	Scale	Depth	Normals	Opacity	Color	Semantics	Mask	
NeuSG	√		√					
GSDF		√	√		√			
3DGSR		√	√		√			
GS-Octree	√			√				
ROGS					√	√		
2DGS		√	√		√			
PGSR	√	√	√		√			
RaDe-GS		√	√		√			
Snap-it					√			
GOF		√	√		√			
High-quality		√	√	√	√		√	
VCR-GauS	√		√		√			
Trim3D			√		√			

Optimization of geometric attributes of 3D Gaussians

3D Gaussians possess geometric attributes such as color, position, and orientation. During training, optimization typically occurs along three dimensions: lighting, depth, and surface normal. Through multiple iterations, 3D Gaussians progressively achieve a better fit to the model surface. GSDF (Yu et al., 2024b) and 3DGSR (Lyu, Wang & Sun, 2024) not only use the lighting loss term from the original 3D GS (Kerbl et al., 2023) but also introduce additional constraints on depth and normal consistency within the loss function. To ensure the computed SDF aligns more closely with the model surface, both methods incorporate an Eikonal penalty term (Li et al., 2024) to regularize the SDF gradient. The distinction between the two lies in GSDF’s use of a curvature loss term to promote surface smoothness, whereas 3DGSR assumes that the SDF value at the Gaussian center approaches zero, thereby forcing Gaussians to adhere closely to the surface. 2DGS (Huang et al., 2024a) and Gaussian Opacity Fields (GOF) (Gropp et al., 2020), which are designed for ray-plane intersection tasks, apply a depth distortion loss term at ray–Gaussian intersection points to minimize the distance between the Gaussians and their corresponding weights. The surface normals of the 3D Gaussians are approximated by the normals of the intersecting planes, thereby improving geometric reconstruction fidelity. Inspired by the alpha blending process, Trim3D (Fan et al., 2024) proposes a novel Gaussian contribution evaluation method and a contribution-based clipping strategy, which further enhances reconstructed surface quality.

3D Gaussians often exhibit scale inconsistencies in multi-view imagery, making it difficult to represent the model surface accurately. Additionally, the centers of 3D Gaussians are typically located inside the surface, which limits their suitability for surface regularization. To address this, NeuSG (Chen, Bao & Wang, 2023) and Octree-GS (Li et al., 2024) propose refining 3D Gaussians into flattened shapes and incorporating a scale regularization term into the loss function, thereby aligning the centers of the flattened Gaussians more closely with the model surface. Lighting plays a critical role in color variation on object surfaces. In multi-view imagery, the same surface point may appear in different colors due to varying camera viewpoints. Unlike VCR-GauS (Chen, Huang & Zhao, 2024), which employs a lighting loss function combining L1 and D-SSIM terms, Planar-based Gaussian Splatting Reconstruction (PGSR) (Chen et al., 2024) introduces photometric multi-view consistency constraints based on planar patches and adds an exposure compensation loss term. These loss terms help adjust the brightness of the rendered image to match the real-world appearance, mitigating the effects of exposure variation across different time frames.

Optimization based on prior knowledge

In addition to imposing constraints on the geometric attributes of 3D Gaussians, some methods incorporate prior knowledge as a form of regularization. Because photometric loss can introduce erroneous surface points, high-quality (Dai et al., 2024) introduces a depth-normal consistency term based on a normal prior in the loss function to improve optimization stability, particularly in regions with strong highlights. R2-Gaussian (Zha, Li & Wang, 2024) implements a voxelizer and its backpropagation in Compute Unified Device Architecture (CUDA), enabling stochastic gradient descent–based optimization of Gaussians. In addition to the photometric loss term, it applies 3D total variation regularization as a homogeneity prior to supporting tomographic 3D reconstruction. To address the challenge of noisy Gaussian point clouds and bridge the gap with smooth 3D surfaces, GS2Mesh (Wolf, Zhang & Feng, 2024) avoids using Gaussian positions directly for depth extraction. Instead, it leverages a pre-trained stereo model and incorporates real-world priors to obtain depth information, thereby improving surface realism and fidelity. GSrec (Wu, Zheng & Cai, 2024) utilizes surface normals and depth as monocular geometric cues to guide the learning of Gaussian position and orientation. In addition, it jointly trains a neural implicit function to approximate both the moving least squares (MLS) function and the normal function derived from the Gaussian distribution. This enables the Gaussians to align more precisely with the surface, resulting in improved mesh quality.

For surface reconstruction in dynamic scenes, the spatial densification and deformation of 3D Gaussians are nonuniform, making it highly challenging to establish a one-to-one correspondence between meshes and 3D Gaussians. To address this, DG-Mesh (Liu, Su & Wang, 2024) incorporates rendering loss, mesh image loss, and L1 loss on rasterization masks into the loss function to jointly optimize mesh geometry and appearance. Additionally, a Laplacian loss is used to promote surface smoothness, while anchor and cycle consistency losses optimize mesh correspondences across frames. The integration of these components improves the performance and coherence of dynamic Gaussian mesh reconstruction. SC-GS (Huang et al., 2024b) adds a lighting loss term to the loss function to support mesh generation under motion variation and incorporates the As-Rigid-As-Possible (ARAP) loss (Sorkine & Alexa, 2007) to avoid local minima and regularize unstructured control points, thereby promoting local rigidity during motion. For target model segmentation, OMEGAS (Wang, Yu & Fang, 2024) addresses the mismatch between lighting loss and object labels by introducing classification and 3D cosine similarity loss terms to improve segmentation consistency.

Optimization for complex scenes

For complex 3D reconstruction tasks involving roads, intricate objects, clothing, and large-scale scenes, optimization strategies are often designed based on scene-specific characteristics. ROGS (Feng et al., 2024) proposes a large-scale road surface reconstruction method using 2D Gaussians. In addition to incorporating color loss, semantic loss, and smoothness terms into the loss function to enhance geometric accuracy, it leverages LiDAR point cloud data to supervise the elevation of Gaussians, ensuring that the reconstructed road surface aligns more closely with real-world elevation profiles. ST2DGS (Wang et al., 2024) introduces a joint optimization of canonical and deformed 2DGS across different timestamps to enable accurate surface reconstruction in dynamic scenes. It also proposes a time-varying opacity model to address occlusion and further improve geometric fidelity. Snap-it (Comi, Sarti & Perotti, 2024) introduces a transmittance loss term applied to contact regions to improve surface reconstruction at points of contact. Additionally, it applies an edge-aware smoothness loss on neighboring masks to reduce uncertainty in other regions and enhance the quality of complex planar reconstructions. GarmentDreamer (Li, Wang & Zhang, 2024) proposes an implicit neural texture field that maps query positions to colors without relying on surface normals or viewing directions. To optimize this field for both reconstruction and refinement, a variation score distillation (Wang, Wu & Liu, 2024) loss term is incorporated into the loss function, resulting in more realistic garment textures. In large-scale scene reconstruction, the simultaneous training of many 3D Gaussians imposes a substantial computational burden. To address this, SA-GS (Xiong, Wang & Zhang, 2024) leverages semantic information to extract the geometric attributes of 3D Gaussians. A semantic loss function is used to guide their shapes toward expected targets. Further optimization of 3D Gaussian shape and attributes is achieved through rendering loss, while a geometric complexity loss and opacity constraint are introduced to reduce Gaussian count, as well as GPU and memory usage during training. The selection of active views during the reconstruction process is critical for optimizing scene quality. Frequency-based (Li, Lajoie & Beltrame, 2024) proposes a view selection method that estimates the potential information gain of candidate viewpoints by transforming rendered images into the frequency domain and selecting the view with the lowest median frequency. This technique exploits blur and artifact patterns in the rendered images to reduce the number of required input views while enhancing reconstruction quality. FisherRF (Jiang, Lei & Daniilidis, 2024) uses Fisher information to guide active view selection and mapping in radiance field models. It quantifies the model’s observational information by computing the Hessian matrix of the loss function and selects views with maximal information gain to improve path planning. Radiance Uncertainty (Ewen et al., 2025) introduces a method for quantifying radiance field model uncertainty through high-order moments of the rendering equation. Modeling the rendering equation as a probabilistic process directly computes pixel-level high-order moments of the model’s output, thereby improving robustness and rendering efficiency.

The optimization of 3D Gaussian sets is fundamental to achieving accurate 3D scene reconstruction. Developing optimization strategies tailored to scene-specific characteristics and incorporating multiple loss constraints can significantly enhance reconstruction quality. However, in practice, excessive constraints may increase computational load and reduce rendering efficiency. Future research should explore integrated optimization frameworks that balance reconstruction fidelity with computational cost.

Surface structure extraction

In 3D reconstruction using 3D Gaussians, highly realistic real-time rendering is achieved by fitting Gaussian points to pixel-wise image contributions when rendering scenes. In addition, the geometric surface of a 3D model can be constructed by leveraging the attributes of the 3D Gaussian set. Depending on whether the scene contains static or dynamic models, surface extraction methods can be categorized into static model surface extraction and deformable model surface extraction.

For static scenes, methods such as Poisson reconstruction (Kazhdan & Hoppe, 2013) and the TSDF method (Curless & Levoy, 1996) are commonly employed to extract surface structures from 3D models represented implicitly within the scene (Lin & Li, 2024; Zhang et al., 2024a; Dai et al., 2024; Wolf, Zhang & Feng, 2024). When Poisson reconstruction is applied, Gaussian noise around the model surface can lead to floating artifacts. To mitigate this, SuGaR (Guédon & Sugar, 2024) introduces a regularization term that constrains 3D Gaussians to fit the surface more tightly. Meshes are subsequently extracted using Poisson reconstruction applied to the rendered depth maps. High-quality (Dai et al., 2024) proposes a volume-cutting strategy that aggregates Gaussian planes with Poisson reconstruction and then applies them to the fused depth maps for surface extraction. OMEGAS (Wang, Yu & Fang, 2024) incorporates semantic information and diffusion models to refine segmentation and enhance geometric detail before applying Poisson reconstruction to extract the surface. While these methods significantly improve reconstruction quality, they also entail long training times and high computational costs.

In addition to leveraging depth information, DN-Splatter (Turkulainen, Wang & Feng, 2024) improves 3D Gaussian attributes by combining priors on depth and surface normals, resulting in better alignment with real-world geometry and enabling more precise reconstruction of indoor scenes. To eliminate dependency on Gaussian normals, GauStudio (Ye, Zhang & Liu, 2024) employs a volumetric fusion method (Vizzo et al., 2022) that integrates median depth values from RGB-D cameras into the mesh, significantly enhancing both reconstruction quality and efficiency. While Poisson reconstruction can become unstable in regions with specular reflections or highlights, adaptive TSDF fusion techniques extract model surfaces in a coarse-to-fine manner, reducing memory usage and improving geometric detail. RaDe-GS (Zhang et al., 2024a) enhances the expressive capacity of 3D Gaussians by applying rasterization techniques to compute depth and normal maps, although TSDF has thus far been applied only to low-resolution voxel grids in large-scale scenes. GS2Mesh (Wolf, Zhang & Feng, 2024) advances TSDF meshing by incorporating stereo matching algorithms to construct RGB-D structures, integrating multiple views into a triangular mesh using TSDF to extract the geometric surface of the mode.

Inspired by neural implicit surface volume rendering techniques, several methods use volume rendering to optimize implicit surfaces. NeuSG (Chen, Bao & Wang, 2023) and 3DGSR (Lyu, Wang & Sun, 2024) employ volume rendering to learn neural implicit surfaces by configuring the attributes and spatial distribution of 3D Gaussians and applying unified optimization and surface constraints. These methods extract surfaces using Poisson reconstruction on Gaussians whose normals are derived from an implicit SDF field. Octree-GS (Li et al., 2024) reconstructs both the SDF and radiance field using volume rendering, encoding them within an octree structure to jointly optimize the SDF and 3D Gaussians. The Marching Cubes algorithm (Lewiner et al., 1987) is then applied to extract the zero isosurface and generate a surface mesh. These approaches simplify the modeling process by optimizing implicit surface representations via volume rendering. However, they are primarily suited for extracting foreground objects (Rosu & Permuto, 2023) and demonstrate low efficiency in capturing fine-grained geometric details and reconstructing background regions. To address this issue, GOF (Yu et al., 2024c) proposes a mesh extraction method based on tetrahedral grids. It determines the contribution of Gaussians to volume rendering by calculating intersection points between rays and Gaussians, evaluating opacity along the ray, and constructing level sets for geometry extraction. In this approach, the centers and corner points of 3D Gaussian bounding boxes are treated as vertices of the tetrahedral grid. After assessing the opacity values of the tetrahedra, the marching tetrahedra algorithm (Shen et al., 2021) is applied to extract the surface mesh. Although GOF enables detailed surface reconstruction for both foreground objects and background regions, 3D Gaussian-based methods continue to suffer from depth estimation errors, which compromise geometric consistency across multiple views.

In dynamic scenes, extracting deformable model surfaces while preserving geometric detail during motion remains a significant challenge. Traditional deformation representations rely on Laplacian coordinates (Sorkine et al., 2004; Sorkine & Alexa, 2007), Poisson equations (Yu et al., 2004), or cage-based techniques (Zhang, Zheng & Cai, 2020). Regularization or data modality constraints are often required to extract deformable surface geometry from dynamic monocular inputs (Yang et al., 2021; Xu & Harada, 2022; Yuan et al., 2022). Three-dimensional (3D) GS has emerged as a powerful approach for surface extraction in dynamic scenes due to its efficient reconstruction quality and real-time rendering capabilities. DG-Mesh (Liu, Su & Wang, 2024) treat 3D Gaussians as oriented point clouds. After generating a mesh using DPSR (Peng et al., 2021) and the Marching Cubes algorithm, the deformable Gaussians are aligned with the mesh surfaces to construct a dynamic Gaussian mesh. GaMeS (Waczyńska et al., 2024) initializes the mesh using neural geometry (Takikawa et al., 2021) or FLAME (Li et al., 2017) and parameterizes the Gaussian components at mesh vertices to align them with the surface. Although Gaussian deformation is achieved by manipulating the mesh, artifacts may emerge when large mesh areas are formed. Mesh-GS (Gao et al., 2024a) employs explicit geometric priors to initialize 3D Gaussians via mesh reconstruction. It guides both segmentation and adaptive refinement of mesh surfaces through Gaussian rendering, while mesh segmentation concurrently informs Gaussian segmentation. These explicit mesh constraints help regularize Gaussian distributions, prevent deformation artifacts, and enable efficient, high-quality reconstruction. GSDeformer (Huang & Yu, 2024) extends cage-based deformation techniques to 3D Gaussians by converting Gaussian representations into occupancy grids and calculating morphological closure. Segmenting contours and constructing a cage mesh achieve simpler and faster deformations than GaMeS, although they do not support real-time performance.

Currently, multiple methods exist for extracting model surface structures based on 3D Gaussian representations. However, due to the inherent limitations in the shape and spatial distribution of 3D Gaussians, the resulting surfaces may exhibit discontinuities or distortions that compromise reconstruction quality. Consequently, a key research challenge in this area is to accurately represent fine geometric features using 3D Gaussians and to minimize errors in surface structure extraction.

Datasets

As graphics technologies evolve and high-precision, high-sensitivity, and high-reliability scanning and detection devices become more accessible, methods for capturing real-world scene data have expanded considerably. This has enabled the acquisition of increasingly detailed datasets from real-world environments. At the same time, synthetic scene data mitigate challenges related to limited or unavailable real-world data, thereby facilitating continuous advancements in 3D reconstruction technologies. Numerous high-quality datasets have been developed through research on 3D model surface reconstruction, supporting applications across indoor, outdoor, and dynamic scenes. Datasets used for surface reconstruction can be broadly categorized into real and synthetic types, depending on how the data are generated. This section introduces the most commonly used datasets for 3D model surface reconstruction, with a summary provided in Table 2.

Table 2 Typical datasets for 3D reconstruction.

Datasets	Number of scenes	Data type	
Tanks and Temples	14	Videos	
Mip-NeRF360	9	Images	
DTU	80	Images	
NeRF-synthetic	8	Images	
D-NeRF	8	Images	
Blendedmvs	113	Images	
Replica	18	Images	
RGBD-SLAM	39	Images	
ScanNet	1,513	Images	
ScanNet++	460	Images	
TUM-RGBD	39	Images	
MtrixCity	2	Images	

Real scene datasets

Real-scene datasets are essential resources in 3D reconstruction research, as they capture complex real-world environments and conditions. These datasets enable researchers to develop more accurate and generalizable models, thereby advancing reconstruction technologies across multiple domains.

The Tanks and Temples dataset (Knapitsch et al., 2017) is a video-sequence-based 3D reconstruction dataset acquired using industrial-grade laser scanners in real-world settings. It comprises 14 scenes, including single-object models such as “Horse” and “Lighthouse,” and large-scale indoor and outdoor environments like “Courtroom” and “Temple.” The dataset supports both camera localization and dense reconstruction tasks, encouraging time-dense sampling to enhance fidelity and realism. The DTU dataset (Aanæs et al., 2016) was captured in a controlled laboratory environment and is a large-scale multi-view stereo dataset with highly accurate camera trajectories. It includes 80 sets of 3D models captured using a structured light scanner and a calibrated camera system. To study the influence of lighting conditions on surface geometry, the dataset provides seven distinct illumination settings, enabling research on how lighting affects multi-view stereo performance, particularly in scenes with reflective or textured surfaces. The Replica dataset (Straub et al., 2019) contains 18 photorealistic 3D indoor scenes composed of dense meshes, high-dynamic-range textures, and rich semantic and instance-level annotations. The scenes were captured using custom-built RGB-D systems and included temporal variations in furniture arrangements, supporting research on semantic mapping, object permanence, and dynamic indoor reconstruction. The RGBD-SLAM Benchmark dataset (Sturm et al., 2012) comprises 39 sequences from two scenarios, including color images, depth images, and detailed camera trajectory information. It introduces two evaluation metrics and associated tools for benchmarking the performance of visual SLAM and odometry systems on RGB-D data.

The ScanNet dataset (Dai et al., 2017) is a large-scale indoor scene dataset that includes semantic annotations for objects and regions within each scene. It supports a range of tasks in 3D scene understanding, including object recognition, semantic segmentation, and scene parsing. Building on this, the ScanNet++ dataset (Yeshwanth et al., 2023) leverages advanced scanning equipment to capture high-precision indoor models. It includes accurate geometric structures, detailed texture information, and spatial relationships between objects. These features support more complex tasks such as object detection, pose estimation, and path planning, enabling robust development and evaluation of 3D scene understanding algorithms. Real-scene datasets offer advantages such as high-fidelity, stable imagery, and accurate laser-based scanning, enabling objective documentation of real-world environments that reflect their physical characteristics and inherent structural regularities. However, they also present limitations, including high data acquisition costs, smaller dataset sizes, and relatively homogeneous scene content. Moreover, because the data are collected in uncontrolled, real-world conditions, ambient lighting variability can introduce noise and reconstruction errors that may adversely affect the accuracy of 3D model generation.

Synthetic scene datasets

Synthetic datasets generated through computer simulations or algorithmic rendering are characterized by their low cost, scalability, and consistently high quality, making them essential in 3D reconstruction research.

The Mip-NeRF360 dataset (Barron et al., 2022) models unbounded 3D scenes using nonlinear scene parametrization, real-time distillation, and orientation distortion regularization. It comprises nine complex scenes featuring intricate objects and backgrounds, enabling multi-view synthesis and detailed depth map generation. This makes it particularly suitable for reconstructing large, complex, and unbounded environments. The Blended-MVS dataset (Yao et al., 2020) includes a wide variety of scenes—ranging from urban areas and buildings to small-scale objects—and provides image sets for training, validation, and testing. It supports novel view synthesis by generating depth maps and color images from unstructured camera trajectories, enhancing the generalization capabilities of 3D reconstruction networks.

The NeRF Synthetic dataset (Mildenhall et al., 2022) contains eight scenes depicting objects such as chairs, drum kits, and hot dogs. Each scene offers 100 images across training, validation, and testing sets, making it well-suited for evaluating deep learning-based reconstruction algorithms on object-centric tasks. The Matrix City dataset (Li et al., 2023a), built with Unity Engine 5, provides high-resolution images of large-scale urban environments. It includes aerial data from two city maps spanning 28 km2, comprising 67,000 aerial views and 452,000 street-level images. This dataset supports configurable environmental parameters—such as lighting, weather, people, and vehicles—and includes depth, normals, and decomposed BRDF materials, making it ideal for city-scale neural rendering and 3D reconstruction research. Synthetic datasets allow the simulation of scenes that are difficult to observe or capture in real-world conditions, thereby enriching dataset diversity and supporting broader generalization scenarios. However, simulations inevitably introduce discrepancies in how real-world phenomena are represented. These inconsistencies can be amplified during model training, leading to reduced generalization performance.

Both synthetic and real-world datasets offer critical support for 3D reconstruction research. Table 3 summarizes the performance metrics of several surface reconstruction methods across three benchmark datasets. Specifically, reconstruction speed was evaluated on the DTU dataset; the F1 score was used to assess accuracy on the Tanks and Temples dataset; and peak signal-to-noise ratio (PSNR) was employed to measure reconstruction quality in dynamic scenes using the D-NeRF dataset (Pumarola et al., 2021). All reported experimental results are derived from original publications, ensuring transparency and reproducibility. While real-world datasets enhance model robustness and realism, their time-consuming collection and annotation processes limit their use for rapid prototyping and algorithm iteration. In contrast, synthetic datasets reduce reliance on manual data acquisition and support large-scale data generation. Each dataset type has distinct advantages and complementary strengths. Joint training with both types can improve performance in diverse 3D reconstruction tasks. Future research will benefit from high-resolution, large-scale, and semantically enriched datasets with fine-grained geometric labels to support increasingly complex reconstruction objectives.

Table 3 Performance metrics of surface reconstruction methods on the benchmark dataset.

Method	DTU	TNT	
37	40	69	105	Barn	Caterpillar	Ignatius	Truck	
NeuS	1.37	0.93	0.57	0.83	0.29	0.29	0.83	0.45	
SuGaR	1.33	1.13	1.15	1.07	0.14	0.16	0.33	0.26	
2DGS	0.91	0.39	0.81	0.76	0.41	0.23	0.51	0.45	
Method	D-NeRF	
	Jumpingjacks	Hook	Hellwarrior	Bouncingballs	
SC-GS	41.13	39.87	42.93	44.91	
DS-Mesh	31.77	27.88	25.46	29.15	
MaGS	42.02	39.80	42.63	42.42	

Applications

This section introduces surface reconstruction tasks based on 3D GS as well as the closely related task of 3D editing (see “3D editing”). Common applications of 3D GS in surface reconstruction fall into three broad categories: autonomous driving, digital human modeling, and large-scale scene reconstruction.

Automatic driving

The applications of 3D GS in autonomous driving are diverse and impactful. Constructing geometric models of roads, buildings, and obstacles is critical for enabling autonomous vehicles to accurately interpret their environments, thereby improving navigation safety and operational reliability. As illustrated in Fig. 2, DriveX (Yang et al., 2024) present a novel framework that integrates generative priors into the reconstruction of the driving scene from single-trajectory recorded videos, addressing limitations in extrapolating novel views beyond a recorded trajectory. Road surface reconstruction based on Gaussian Splatting (RoGS) (Feng et al., 2024) uses 2D Gaussians to reconstruct road surfaces at large scale. These reconstructions support lane detection and automatic annotation, contributing to more precise path planning in real-time scenarios. TCLC-GS (Zhao, Huang & Fang, 2024) combines the strengths of LiDAR and camera sensors by aligning 3D Gaussians with mesh structures, generating rich geometric and color representations of urban street scenes. This integration enhances environmental perception and accelerates real-time rendering, thereby improving the situational awareness and safety of autonomous systems. DrivingForward (Tian et al., 2024) introduces a feed-forward GS framework that jointly trains a pose estimation network, a depth prediction network, and a Gaussian rendering network. This system predicts both the appearance and position of 3D Gaussians from surround-view inputs, enabling the reconstruction of dynamic driving environments from flexible viewpoints.

Figure 2 Visualization of road reconstruction.

Source: Yang et al. (2024); https://doi.org/10.48550/arXiv.2412.01717.

Digital man

The use of digital human models spans diverse domains, including education, entertainment, and virtual reality. GauFace (Qin et al., 2024) employs geometric priors and constrained optimization to construct a clean and structured Gaussian representation, enabling high-fidelity, real-time facial interactions on platforms such as computers and mobile devices. This method supports efficient animation and physics-based rendering of facial assets. MoSca (Lei, Wu & He, 2024) develops animatable digital human avatars from both multi-view and monocular video inputs. By initializing 3D Gaussians on mesh surfaces, the system reconstructs detailed geometric models of the human body, achieving realistic lighting and interactive rendering, thereby extending the application of digital humans in virtual environments. As illustrated in Fig. 3, the proposed framework (Nguyen et al., 2025) synergistically combines 3D Gaussian Splatting with adaptive light-field shading to achieve real-time photorealistic rendering of dynamic facial animations, while maintaining sub-millisecond latency through optimized neural computation pipelines. MeshAvatar (Chen et al., 2025) extracts explicit triangular meshes from implicit SDF fields to represent digital avatars. It combines this geometry with physics-based rendering to decompose and reconstruct both shape and texture components, enabling flexible editing and manipulation of digital human models.

Figure 3 Editing and manipulation of digital human models.

Source: Nguyen et al. (2025); https://doi.org/10.48550/arXiv.2502.08085.

Large-scale scene reconstruction

Large-scale scene reconstruction is essential for applications such as digital city modeling, road planning, and immersive virtual environments. These use cases require not only photorealistic rendering but also high-precision geometric modeling. Real-time Gaussian SLAM (RTG-SLAM) (Peng et al., 2024) introduces a method that utilizes both transparent and opaque 3D Gaussians to reduce the total number of Gaussians needed for accurate surface representation. This approach improves the realism of rendered scenes without compromising geometric fidelity and enables real-time reconstruction of diverse real-world environments, including corridors, storerooms, residences, and offices. As shown in Fig. 4, Vast Gaussian (Lin et al., 2024) adopts a divide-and-conquer strategy to partition large-scale scenes into smaller spatial cells, distributing training views and point clouds across these subregions. This method supports parallel optimization and seamless merging, enabling efficient 3D reconstruction and real-time rendering of complex environments such as urban structures, water networks, and transportation infrastructure. In contrast to techniques that rely on Structure-from-Motion (SfM) for initial point cloud generation, Gaussian Pro (Cheng et al., 2024) reconstructs scenes using predefined geometric shapes and applies block matching to initialize 3D Gaussians with precise positions and orientations. This method is particularly effective for optimizing large-scale reconstructions involving untextured or low-textured surfaces.

Figure 4 3D reconstruction of an urban scene (Lin et al., 2024).

3D editing

3D editing refers to the reconstruction, modification, and refinement of 3D models and is widely applied in domains such as game development, film production, industrial design, and architectural planning. Editing and optimizing 3D models enhance their reusability and adaptability across varied applications. Direct (Lin & Li, 2024) introduces a differentiable marching algorithm to learn a mesh representation from which 3D Gaussians are generated. By manipulating this mesh, users can flexibly control both the appearance and geometry of 3D models. SC-GS (Huang et al., 2024b) enables motion editing by adjusting sparse control points, allowing for rigid pose deformations while preserving high-fidelity rendering quality. As illustrated in Fig. 5, Mani-GS (Gao et al., 2024b) proposes an adaptive triangular shape-aware Gaussian binding method that supports large-scale deformations, localized adjustments, and soft-body simulations for enhanced model expressiveness. Inspired by 3D GS techniques, Tetrahedron (Gu et al., 2024) introduces tetrahedron splatting for AI-driven modeling, enabling the generation and manipulation of 3D content from text prompts. This approach significantly broadens the creative and functional scope of 3D editing workflows.

Figure 5 Principle of the Mani-GS method (Gao et al., 2024b).

Challenges

This article introduces the core concepts of 3D GS and systematically reviews surface reconstruction methods based on 3D GS, covering technological foundations, datasets, and practical applications. Despite these advancements, a performance gap persists between reconstructed surfaces and actual model geometries. Furthermore, improvements are needed in areas such as dataset quality and training optimization. The primary challenges in this field can be summarized as follows: (1) Building higher-quality datasets: Existing datasets for 3D modeling are limited in both quantity and quality. Laboratory datasets, though precise, lack diversity, are expensive to acquire and often omit key variables such as lighting and material properties that affect real-world appearance. In contrast, synthetic datasets typically focus on object geometry while neglecting environmental context. These discrepancies can accumulate during training and degrade the generalization ability of reconstruction algorithms. Addressing this challenge requires the development of large-scale, fine-grained 3D datasets that capture both scene complexity and object variation across lighting and material conditions.

(2) Batch extraction in large-scale scenes: Techniques designed for small-scale or single-object reconstructions often perform poorly when scaled to large environments. Regularization across vast numbers of 3D Gaussians imposes high computational costs and diminishes efficiency. To resolve this, there is a need to restructure how 3D Gaussians are organized and to design intelligent batch modeling strategies that ensure both semantic consistency and geometric coherence. Such strategies are especially critical for efficient and scalable reconstruction of urban environments.

(3) Improving the quality of the geometric details on model surfaces: Several factors affect the accuracy of surface geometry extraction. Current methods typically rely on fixed-resolution training images, which can result in artifacts and loss of fine details when the resolution is reduced or non-optimal. To improve reconstruction fidelity, it is necessary to support variable-resolution training pipelines and integrate image upscaling and parameter optimization. This would enable the generation of high-quality surface models with finer geometric structures, even under suboptimal imaging conditions.

Conclusions

3D reconstruction has a long-standing research history and continues to evolve across disciplines. In recent years, methods based on 3D GS have achieved significant progress in improving both geometric accuracy and visual fidelity. This study offers a comprehensive summary of surface reconstruction techniques grounded in 3D GS, covering core technologies, dataset development, and emerging applications. It also identifies current limitations and discusses prospective solutions. Looking ahead, ongoing improvements in 3D GS-based reconstruction methods are expected to enable the creation of more geometrically accurate and semantically complete 3D models. This progression drives the transformation and advancement of 3D reconstruction technologies.

We would like to thank Editage for English language editing.

Additional Information and Declarations

Competing Interests

The authors declare that they have no competing interests.

Author Contributions

Zheng Xu conceived and designed the experiments, performed the experiments, analyzed the data, prepared figures and/or tables, and approved the final draft.

Gang Chen conceived and designed the experiments, performed the experiments, authored or reviewed drafts of the article, and approved the final draft.

Feng Li conceived and designed the experiments, analyzed the data, prepared figures and/or tables, authored or reviewed drafts of the article, and approved the final draft.

Lingyu Chen performed the experiments, prepared figures and/or tables, authored or reviewed drafts of the article, and approved the final draft.

Yuanhang Cheng analyzed the data, authored or reviewed drafts of the article, and approved the final draft.

Data Availability

The following information was supplied regarding data availability:

This is a literature review.

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
