# Peer review of "A survey on surface reconstruction based on 3D Gaussian splatting"

_PeerJ Computer Science, doi:10.7717/peerj-cs.3034_

## Round 0.1 · original submission · Major Revisions

Dear authors,
You are advised to critically respond to all comments point by point when preparing an updated version of the manuscript and while preparing for the rebuttal letter. Please address all comments/suggestions provided by reviewers, considering that these should be added to the new version of the manuscript.

Kind regards,
PCoelho

Reviewer 1 ·

Basic reporting

The paper primarily reviews the use of Gaussian Splatting for surface reconstruction, covering most of the current mainstream directions and methods in the field. It provides a comprehensive overview of existing approaches, making it a valuable resource for researchers interested in this topic.

However, I noticed that the formatting of the equations appears somewhat unusual, as they are left-aligned. I am unsure whether this is due to the journal's formatting requirements or another reason, but it might be worth checking to ensure consistency and readability.

Additionally, while the paper lists various methods, it would be beneficial to include their performance metrics on standard benchmark datasets such as DTU and TNT. This would make it easier for readers to compare different approaches and better understand their effectiveness in practical scenarios.

Experimental design

-

Validity of the findings

-

Cite this review as

Reviewer 2 ·

Basic reporting

This survey paper provides a comprehensive overview of surface reconstruction techniques based on 3D Gaussian Splatting (3DGS), an emerging approach in radiance field modeling and 3D representation. The paper begins by outlining the background of 3DGS and then systematically examines three key components of the reconstruction process:

1. 3D Representation:
The survey categorizes methods into those for static and dynamic scenes. It discusses various representation schemes, including implicit models (e.g., Signed Distance Functions) and explicit models (e.g., point clouds), highlighting their respective strengths and challenges.

2. Model Optimization:
The paper reviews three distinct optimization strategies. One notable approach involves refining the geometry attributes obtained from 3DGS by incorporating additional priors. This strategy is particularly effective in enhancing geometric accuracy and addressing the complexities inherent in large-scale scenes.

3. Surface Extraction:
The survey describes a common methodology for extracting surfaces from the 3D representations. This step is crucial as it bridges the gap between abstract 3D models and practical, usable surfaces.

Finally, the paper concludes with an overview of several public datasets that serve as benchmarks for evaluating surface reconstruction techniques. The author also mentioned several applications and challenges in the community. Overall, the survey not only maps the current landscape of 3DGS-based surface reconstruction but also identifies key areas for future research.

Experimental design

The surface extraction section of the survey can be enhanced by a detailed exploration of classical surface extraction methods, which form the backbone for many modern reconstruction techniques.

1. Marching Cubes (MC):
The survey now begins with an in-depth discussion of the Marching Cubes algorithm—a fundamental method for converting volumetric data into polygonal surface representations. MC subdivides the volume into a grid of cubes and determines the intersection of an isosurface with each cube's edges through linear interpolation. The MC algorithm is not only a standalone method but also forms a crucial component in more sophisticated approaches, such as Poisson Surface Reconstruction and TSDF volume-based methods, which leverage its principles to achieve improved surface accuracy and robustness.

2. Advancements Beyond MC:
Building upon the foundation laid by Marching Cubes, most methods like TSDF volume and Poisson Surface reconstruction have been developed and used in recent 3DGS methods. Similarly, the Marching Tetrahedra algorithm is presented as an evolution of MC. This method, used in frameworks like GOF, resolves ambiguities inherent in the cube-based approach by subdividing the volume into tetrahedra, thereby yielding more consistent surface topology, especially in complex geometries.

By detailing these classical methods along with their corresponding equations and subsequent advancements, the survey provides a comprehensive summary that not only clarifies the fundamental algorithms but also highlights their evolution within the context of 3D Gaussian Splatting-based surface reconstruction.

Validity of the findings

The overall survey is comprehensive and vital for the community. It lists most of the key works in this area. The author also proposes some challenges, which are indeed some crucial problems.

Additional comments

Here are some missing references that could further improve the survey.
For static 3DGS surface reconstruction:
a) Surface Reconstruction from 3D Gaussian Splatting via Local Structural Hints
b) Trim 3D Gaussian Splatting for Accurate Geometry Representation
c) Quadratic Gaussian Splatting for Efficient and Detailed Surface Reconstruction
For dynamic 3DGS surface reconstruction:
d) Space-time 2D Gaussian Splatting for Accurate Surface Reconstruction under Complex Dynamic Scenes

Here are some typos in the draft
1. The authors of 2DGS are wrong.1
2. GS-octree should be Octree-GS

Cite this review as

Reviewer 4 ·

Basic reporting

This survey on surface reconstruction based on 3D Gaussian splatting is of broad and cross-disciplinary interests: wide applications in computer graphics, such as animation, virtual reality, and game design. Surface reconstruction methods are applicable in diverse domains, including GIS, medical imaging, robotics, and cultural heritage.
There are some reviews on 3D Gaussian reconstruction. However, the field of computer graphics and 3D reconstruction is evolving quickly, particularly with the introduction of new algorithms and methodologies like 3D Gaussian splatting, so this review gives a more updated literature in this topic.
The introduction appears to adequately introduce the subject of surface reconstruction based on 3D Gaussian splatting. The introduction provides context by mentioning the foundational nature of surface reconstruction in computer graphics and its growing significance due to advancements in neural radiance fields (NeRFs) and 3D Gaussian splatting. It clearly states the aim of the survey: to present core methodologies for 3D model surface reconstruction and to establish a structured roadmap encompassing various aspects of the field. This sets clear expectations for the reader. The introduction outlines the main components that will be discussed, including 3D representations, reconstruction methods, datasets, applications, challenges, and future research directions.

Experimental design

The survey methodology appears to be structured to offer a comprehensive and unbiased coverage of the subject of surface reconstruction based on 3D Gaussian splatting.
Here are some points to consider regarding its consistency and potential areas for improvement: Including empirical studies or case examples that demonstrate the effectiveness of the discussed methods would strengthen the survey. This could involve comparing performance metrics across various algorithms. Also, the literature should shortly include reconstruction work in 3D-GS related SLAM (https://arxiv.org/abs/2312.06741, https://arxiv.org/abs/2312.02126, https://arxiv.org/abs/2312.10070) and active view selection in reconstruction (https://arxiv.org/abs/2311.17874, https://arxiv.org/abs/2503.14665, https://arxiv.org/abs/2409.16470)
Regarding the dataset part, some datasets should be added: https://ieeexplore.ieee.org/document/6385773, https://arxiv.org/abs/1702.04405, https://arxiv.org/abs/2308.11417, as there are already some 3D GS-based reconstruction works in SLAM using these datasets.
Overall, the survey seems to be organized logically into coherent paragraphs and subsections, facilitating a clear understanding of the subject matter.

Validity of the findings

The survey has a well-developed and supported argument that aligns with the goals set out in the introduction. The introduction outlines the aim of the survey: to present core methodologies for 3D model surface reconstruction and establish a structured roadmap. The subsequent sections seem to fulfill this by addressing these components systematically. The review discusses various aspects of surface reconstruction, including 3D representations, reconstruction methods, datasets, applications, and challenges. This comprehensive approach supports the goal of providing a thorough understanding of the subject. By addressing ongoing challenges and suggesting future research directions, the survey meets its goal of not only summarizing current knowledge but also guiding future inquiry. This forward-looking perspective is crucial for a well-rounded argument.
The conclusion of the survey appears to effectively identify unresolved questions, gaps, and future directions in the field of surface reconstruction based on 3D Gaussian splatting. By acknowledging challenges and suggesting areas for further research, it reinforces the importance of ongoing inquiry and innovation in the field.

Additional comments

Apart from adding the literature suggested above, there is one minor rectification to be made: the authors should change SFM to Structure from Motion(SfM) as this is the common usage of the term in existing literature.

Cite this review as

---

## Round 0.2 · Minor Revisions

Dear authors,

Thanks a lot for your efforts to improve the manuscript.

Nevertheless, some concerns are still remaining that need to be addressed.

Like before, you are advised to critically respond to the remaining comments point by point when preparing a new version of the manuscript and while preparing for the rebuttal letter.

Kind regards,
PCoelho

Reviewer 1 ·

Basic reporting

The manuscript is clearly written in professional and unambiguous English. The introduction provides sufficient background and clearly states the motivation and relevance of the review. The literature is thoroughly referenced, with coverage of both foundational and recent work (including 2023–2024 papers), ensuring the review is timely and comprehensive. The article structure is logical and consistent with the norms of literature reviews, including sections on background, key technologies, datasets, applications, and challenges. Figures and tables are well-designed and support the text effectively.

Experimental design

This is a well-scoped literature review focused on 3D Gaussian Splatting (3D GS) for surface reconstruction—a timely and important topic in computer graphics and vision. The authors present a structured overview of the field, categorizing recent methods into scene representation, optimization strategies, and surface extraction. The methodology for literature collection is transparent, with searches conducted across CVF, arXiv, and Web of Science using relevant keywords. The paper is well-organized and succeeds in giving readers a coherent and systematic understanding of the topic.

Validity of the findings

The review presents a coherent and well-supported analysis of existing work, with conclusions clearly linked to the objectives set out in the introduction. The discussion of future challenges—such as dataset quality, geometric detail extraction, and large-scale reconstruction efficiency—is particularly insightful and constructive. The authors have strengthened the final sections during the revision phase, making the conclusions more balanced and forward-looking. Overall, the findings and categorizations are valid, well-argued, and of practical value to researchers in the field.

Additional comments

The authors have adequately addressed all concerns raised in the previous review round, demonstrating a careful and thoughtful revision process.

Compared to existing surveys, this manuscript offers a unique perspective by centering the discussion specifically on surface reconstruction with 3D Gaussian Splatting, which fills an important gap in the current literature.

I recommend minor proofreading before final submission to polish residual phrasing issues, although the current version already meets the standard for publication.

Cite this review as

Reviewer 4 ·

Basic reporting

The draft fulfills the requirements of all areas in basic reporting.

Experimental design

The Survey Methodology appears to be briefly described. The authors searched three electronic databases—Computer Vision Foundation Open Access Database, arXiv.org, and Web of Science—using specific keywords related to 3D Gaussian splatting, surface reconstruction, and related topics. While this indicates an organized approach to collecting relevant literature, the description lacks detailed information about the criteria used for selecting and filtering studies, the time frame considered (beyond mentioning recent preprints from 2024), and whether any measures were taken to ensure unbiased coverage of all significant approaches. To enhance its comprehensiveness and reduce potential bias, additional details such as the selection process, screening procedures, manual searches for important works not found via keyword searches, and steps to include various viewpoints or approaches would be helpful.

Validity of the findings

The draft fulfills the requirements of all areas in validity of the findings.

Additional comments

no comment

Cite this review as

---

## Round 0.3 · accepted · Accept

Dear authors, we are pleased to verify that you meet the reviewer's valuable feedback to improve your research.

Thank you for considering PeerJ Computer Science and submitting your work.

Kind regards
PCoelho

Reviewer 4 ·

Basic reporting

The draft fulfills the requirements of all areas in basic reporting.

Experimental design

The Survey Methodology is consistent with a comprehensive, unbiased coverage of the subject.
The sources are adequately cited.
The review organized logically into coherent paragraphs/subsections.

Validity of the findings

The draft fulfills the requirements of all areas in validity of the findings.

Additional comments

no comment

Cite this review as